Leopard and spotted hyena densities in the Lake Mburo National Park, southwestern Uganda

Braczkowski Aleksander 1 2 3 braczkowski@sustech.edu.cn
Schenk Ralph 4
http://orcid.org/0000-0003-3482-799X Samarasinghe Dinal 5
Biggs Duan 2 6 7
Richardson Allie 8
Swanson Nicholas 4
Swanson Merlin 4
http://orcid.org/0000-0003-4193-7150 Dheer Arjun 9
Fattebert Julien 10 11
1 School of Environmental Science and Engineering , Southern University of Science and Technology , Shenzhen, China
2 Resilient Conservation Group, Centre for Planetary Health and Food Security, Griffith University , Nathan, Queensland , Australia
3 School of Natural Resource Management, Nelson Mandela University, George Campus , George, Western Cape , South Africa
4 Mihingo Lodge , Kampala , Uganda
5 Wildlife Research and Nature Conservation Foundation (WRNCF) , Colombo , Sri Lanka
6 School of Earth and Sustainability. Northern Arizona University , Flagstaff, Az , USA
7 Centre for Complex Systems in Transition, School of Public Leadership , Stellenbosch University , Stellenbosch, South Africa
8 School of Biological Science, The University of Queensland , Brisbane , Queensland
9 Department of Evolutionary Ecology, Leibniz Institute for Zoo and Wildlife Research , Berlin , Germany
10 Wyoming Cooperative Fish and Wildlife Research Unit, Department of Zoology and Physiology, University of Wyoming , Laramie, Wyoming , United States
11 Centre for Functional Biodiversity, School of Life Sciences, University of KwaZulu-Natal , Durban, KwaZulu-Natal , South Africa
Schuster Richard
Electronic publication date: 2022 Jan 27
Publication date: 2022
Volume: 10
Electronic Location ID: e12307
Received 2021 Mar 16; Accepted 2021 Sep 22
Copyright: © 2022 Braczkowski et al.
Copyright year: 2022
Copyright holder: Braczkowski et al.
License: This is an open access article distributed under the terms of the Creative Commons Attribution License, which permits unrestricted use, distribution, reproduction and adaptation in any medium and for any purpose provided that it is properly attributed. For attribution, the original author(s), title, publication source (PeerJ) and either DOI or URL of the article must be cited.
License URL: https://creativecommons.org/licenses/by/4.0/

Keywords: Panthera pardus, Crocuta crocuta, Spatially explicit capture-recapture, Population size, East Africa, Human-carnivore conflict

Funding: Scientific Exploration Society Rufford Foundation Mihingo Lodge Siemiatkowski Foundation The Scientific Exploration Society, Rufford Foundation, Mihingo Lodge, and the Siemiatkowski Foundation funded Alex Braczkowski while he was in the field. The funders had no role in study design, data collection and analysis, decision to publish, or preparation of the manuscript.

==============================
Robust measures of animal densities are necessary for effective wildlife management. Leopards (Panthera pardus) and spotted hyenas (Crocuta Crocuta) are higher order predators that are data deficient across much of their East African range and in Uganda, excepting for one peer-reviewed study on hyenas, there are presently no credible population estimates for these species. A lack of information on the population status and even baseline densities of these species has ramifications as leopards are drawcards for the photo-tourism industry, and along with hyenas are often responsible for livestock depredations from pastoralist communities. Leopards are also sometimes hunted for sport. Establishing baseline density estimates for these species is urgently needed not only for population monitoring purposes, but in the design of sustainable management offtakes, and in assessing certain conservation interventions like financial compensation for livestock depredation. Accordingly, we ran a single-season survey of these carnivores in the Lake Mburo National Park of south-western Uganda using 60 remote camera traps distributed in a paired format at 30 locations. We analysed hyena and leopard detections under a Bayesian spatially explicit capture-recapture (SECR) modelling framework to estimate their densities. This small national park (370 km2) is surrounded by Bahima pastoralist communities with high densities of cattle on the park edge (with regular park incursions). Leopard densities were estimated at 6.31 individuals/100 km2 (posterior SD = 1.47, 95% CI [3.75–9.20]), and spotted hyena densities were 10.99 individuals/100 km2, but with wide confidence intervals (posterior SD = 3.35, 95% CI [5.63–17.37]). Leopard and spotted hyena abundance within the boundaries of the national park were 24.87 (posterior SD 7.78) and 39.07 individuals (posterior = SD 13.51) respectively. Leopard densities were on the middle end of SECR studies published in the peer-reviewed literature over the last 5 years while spotted hyena densities were some of the first reported in the literature using SECR, and similar to a study in Botswana which reported 11.80 spotted hyenas/100 km2. Densities were not noticeably lower at the park edge, and in the southwest of our study site, despite repeated cattle incursions into these areas. We postulate that the relatively high densities of both species in the region could be owed to impala Aepyceros melampus densities ranging from 16.6–25.6 impala/km2. Another, potential explanatory variable (albeit a speculative one) is the absence of interspecific competition from African lions (Panthera leo), which became functionally extinct (there is only one male lion present) in the park nearly two decades ago. This study provides the first robust population estimate of these species anywhere in Uganda and suggests leopards and spotted hyenas continue to persist in the highly modified landscape of Lake Mburo National Park.

Introduction

Precise measures of animal densities represent one of the most fundamental precursors for effective wildlife management (Karanth, 1995; White & Burnham, 1999; Duangchantrasiri et al., 2016; Rayan & Linkie, 2015). Density estimates assist inter alia with species assessments (Jacobson et al., 2016), the setting of harvest quotas (Balme, Slotow & Hunter, 2009), and in gauging the viability of individual populations (Sollmann et al., 2011). Measures of animal abundance and density are becoming increasingly critical for species that are exposed to significant anthropogenic pressures, are constrained to small habitat patches, and are important to the economies of developing nations (O’Bryan et al., 2018).

Large carnivores naturally occur at relatively low densities and have large space requirements (Balme, Slotow & Hunter, 2009; Gopalaswamy et al., 2012b). Anthropogenic sources of mortality at the edges of small reserves can therefore depress carnivore densities, even within protected areas because animals move beyond their boundaries and are killed (e.g., Balme, Slotow & Hunter, 2009; Woodroffe & Ginsberg, 1998). In Uganda, most protected areas are relatively small, isolated and have high human pressures at their edges (Venter et al., 2016). Additionally, most Ugandan national parks and wildlife reserves are bordered by livestock rearing communities, and large carnivores regularly kill livestock in these areas (Ochieng, Ahebwa & Visseren-Hamakers, 2015). Consequently, large carnivores are often killed in retaliation for livestock killing, and damage through poisoning, trapping or shooting (Tweheyo et al., 2012).

Leopard (Panthera pardus) and spotted hyena (Crocuta crocuta) are examples of species which have impacts on the livelihoods of local communities in Uganda (Ochieng, Ahebwa & Visseren-Hamakers, 2015). Both species were responsible for 1,102 attacks on cattle, sheep and goats on the edge of Lake Mburo National Park (hereafter LMNP) (spotted hyenas n = 762 or 69%, leopards n = 340 or 31% between January 2009–December 2018, Braczkowski et al., 2020c). Such conflict between these species and pastoralists may have ramifications and at least 19 leopards were killed on the boundary of LMNP in a 4-year period from 2003–2006 (CITES CoP 14 Proposal 3), and two hyena clans (each >14 individuals in size) that were regularly viewed by tourists were poisoned in 2007 (R Schenk, 2018, personal communication). However, both species are also important for the wildlife-viewing tourism (Van der Meer, Badza & Ndhlovu, 2016) and in Uganda in 2018 alone, 1,585 people purchased a night game drive permit for leopard viewing in LMNP, equating to US$47,550 in revenue for the Ugandan Wildlife Authority (A Kule, 2018, personal communication). This often leads to contradictory management goals, where one entity seeks higher densities to maximize tourism revenue, and the other seeks lower densities due to livelihood losses accrued from conflict. However, a lack of robust information on the population status of leopards and spotted hyenas inhibits the design of sustainable management offtakes and also in assessing the impact of conservation interventions on carnivores and communities (e.g., Financial compensation, the erection of livestock protection bomas etc.).

To address these concerns, we sought to estimate the population abundance and densities of leopards and spotted hyenas in the LMNP, south-western Uganda. LMNP is a small, protected area that lacks much of the charismatic megafauna found elsewhere in the country e.g., mountain gorillas (Gorilla beringei beringei), chimpanzees (Pan troglodytes), African elephants (Loxodonta africana) and lions (Panthera leo). Consequently, leopards and spotted hyenas are important tourism draws for the region. This is even more important as African lions became functionally extinct in LMNP in the early 2000’s (Uganda Wildlife Authority, 2010). There is also legal trophy hunting of leopards on LMNP’s edge and high rates of human-leopard conflict on its boundary (Braczkowski et al., 2020c). This study represents the first assessment of leopards undertaken in a protected area system in Uganda and provides one of the first spatially explicit estimates of spotted hyena densities in the literature. This study produces a baseline single season snapshot into the population densities for both species to inform conservation management in the region and to better track the impacts of conservation interventions.

Study area

We studied leopards and spotted hyenas in the LMNP (370 km2), Kiruhura district, Western Uganda (30°47′–31°04′E, 00°30′–0°30′S, Fig. 1). The LMNP forms part of the Akagera savanna ecosystem which extends from Rwanda and north-western Tanzania down into south-western Uganda (Menaut, 1983; Van de Weghe, 1990). LMNP experiences a bimodal annual rainfall pattern (October–December and February–June) and annual rainfall and temperatures average 800 mm and 28 °C, respectively (Moe et al., 2016). The woody vegetation in the park is characterized by dry Acacia savanna dominated by Acacia hockii, woodlands, thickets and swamps which occur on the edges of Lake Kachera and Mburo (Rannestad et al., 2006). The most common grasses include (Loudetia kagerensis), (Chloris gayana), and (Sporobolus pyramidalis). LMNP supports one of two remaining population of impala (Aepyceros melampus) in Uganda, the most common and preferred prey of the African leopard (Hayward et al., 2006). The park also harbours Plains zebra (Equus quagga), Cape buffalo (Syncerus caffer), Defassa waterbuck (Kobus ellipsiprymnus defassa), bushbuck (Tragelaphus scriptus) and warthog (Phacochoerus africanus, Rannestad et al., 2006). There is only one male lion (≥10 years old) in LMNP (a vagrant thought to have come from Akagera National Park, in neighbouring Rwanda). LMNP is bordered by a matrix of small human settlements, small-scale subsistence crops, dairy ranches and communal grazing lands (Ochieng, Ahebwa & Visseren-Hamakers, 2015).

Figure 1 Study area map of the Lake Mburo National Park.

Park history, introduction of trophy hunting and human-carnivore conflict

Although the national park itself is small, much of the former park area—which is now mainly used as cattle rangeland-still has considerable woodlands, thickets and natural vegetation and Rannestad et al. (2006) noted higher densities of bushbuck, impala, reedbuck (Redunca redunca), waterbuck and zebra outside of the national park’s borders during the wet season. The region surrounding LMNP has a trophy hunting scheme which was initiated due to increasing complaints by communities, stating that the increasing wildlife was a nuisance (Ochieng, Ahebwa & Visseren-Hamakers, 2015). The leopard is only allowed to be hunted when a problem animal tag is made available by Ugandan Wildlife Authority (hereafter UWA) attributed to repeated stock killing and damage. Although harvests of leopards since 2007 have been low in Uganda (17 skins, skulls and trophies exported from 2009–2017), attempts were made to have the species downgraded from CITES Appendix 1 to Appendix 2 and proposed a quota of 50 leopards annually (despite the lack of even a single abundance estimate anywhere in the country, CITES CoP 14 Proposal 3). Currently, 28 leopards are available annually on quota country-wide. Contrastingly, in Africa, hyenas are often taken opportunistically by trophy hunters rather than as prized trophy animals and we could not find any evidence that they are an actively hunted species in Uganda (see for example: http://www.uganda-wildlife-safaris.com).

Methods

Camera trapping

This research was granted approval by the Uganda Wildlife Authority under permit number: UWA/COD/96/05 as approved by the Executive Director Mr Stephen Masaba. We implemented one single season camera-trap survey for 53 days in the LMNP from 26 July 2018–16 September 2018 using Cuddeback™ 20-megapixel Long Range IR camera traps (powered by 8 AA batteries each) set in a paired format. The survey encompassed 30 camera trap sites distributed across the national park (Fig. 1), but we omitted camera traps in the far western sector of the park due to a lack of road access. Each camera trap site consisted of two camera traps, each mounted to a 1 m steel pole 40 cm from the ground. We positioned each camera perpendicular to a vehicle track or game trail at a 60–75°-angle to facilitate early detection of leopards and spotted hyenas. We set our camera traps on roads, vehicle tracks, trails and drainage lines, as these are regularly used by leopards and spotted hyenas as travel and hunting routes (Balme, Hunter & Slotow, 2009; Balme, Slotow & Hunter, 2009; Henschel, Malanda & Hunter, 2014). We checked traps every 4–7 days to correct for animal damage, replace memory cards and to assess battery functionality (Braczkowski et al., 2016). Camera traps were set to burst mode and took five images every time the infrared sensor was triggered. We set camera traps in a way as to ensure that at least one camera-trap site was present in an area corresponding to the smallest female leopard home-range recorded in the literature (30 km2; Bailey, 1993, and 23 km2 in Fattebert et al., 2016), as these are smaller than male leopards and spotted hyenas. Our camera spacing was 2.1 km (5–7 camera stations per female home range). We chose this camera spacing in order to ensure that no animal had a zero probability of capture (Karanth & Nichols, 1998). The identity of individual leopards and spotted hyenas was determined by their unique rosette and spot patterns (Miththapala et al., 1989; O’Brien & Kinnaird, 2011). For leopards, we were able to classify the sex of individuals by using distinctive morphological cues such as the presence of testes and the enlarged dewlap and sagittal crest in males (Balme, Hunter & Braczkowski, 2012; Braczkowski et al., 2015a).

The first and third author assigned individual identity to temporally unique photographs and only included into the final density estimation process individuals for which there was consensus (Bahaa-el-din et al., 2016). We excluded images that were blurred, were too far away from the camera trap and those where observers could not agree on identity. For the purpose of building capture histories with known unique individual identities, we used both flanks of leopards in our analysis (Fig. 2). Spotted hyenas, however, often walked around cameras and did not present a clear flank on both sides of a single animal, and several individuals moved around a single camera at the same time. To avoid mismatching flanks and mistakenly double-count individuals, we chose the flank of hyenas with the highest number of photographs recorded during our survey (Henschel, Malanda & Hunter, 2014).

Figure 2 Individual identification of spotted hyenas and leopards from camera traps.

Individual identification information extracted from leopards and spotted hyenas in the LMNP, 2018. Slide 1 (top) denotes a female leopard captured at trap location five on sampling occasion two and 10 respectively. Rosette patterns and facial spots were extracted during these two occasions. Slide 2 (bottom) denotes the spot pattern extracted from a spotted hyena captured at location 12 and 27 on sampling occasions 5 and 22 respectively.

SECR modelling

We estimated leopard and spotted hyena densities and abundance in LMNP using Bayesian spatially explicit capture-recapture modelling. By incorporating spatial information into the detection process, the method does not suffer from the “edge effects” common to non-spatial estimators (Gopalaswamy et al., 2012a). The modelling approach uses a state (leopard and spotted hyena population size and locations in the landscape) and observation process (Royle et al., 2009; Gopalaswamy et al., 2012b). To accurately estimate the densities and home-range centres of both species we generated potential activity centres across our study area (370 km2) in the form of 0.336 km2 (i.e., 580 m × 580 m, Gopalaswamy et al., 2012a) equally spaced pixels. This state-space assumes the number of leopards and spotted hyenas found in these pixels are defined by a binomial process, but because spotted hyenas are often found in groups, the state process allows for ≥2 spotted hyenas to have an activity centre in the same pixel (Gopalaswamy et al., 2012a). The state space encompassed the LMNP, and a buffer of 25 km around it (including the eastern rangelands bordering the park, Kanyaryeru and the southern farmlands). We masked out all human settlements and water bodies inside and surrounding the national park, as leopards and spotted hyenas are unlikely to have their home-range centres directly in such unsuitable habitats (Royle et al., 2009; Gopalaswamy et al., 2012a; Braczkowski et al., 2016). We used a classical capture re-capture sampling design and created a standard capture re-capture matrix (trap locations, individual leopards or hyenas and sampling occasions, e.g., du Preez, Loveridge & Macdonald, 2014; Braczkowski et al., 2016; Williams et al., 2017). Large terrestrial carnivores regularly feature differences at the sex-level in their home-range sizes and capture probability (Gopalaswamy et al., 2012b; Braczkowski et al., 2016). Differences in movements of animals based upon sex can affect the observation process in spatial capture-recapture (Sollmann et al., 2011). To factor this into our models, we included a sex-specific covariate in the observation process and accounted for different capture probability for leopards. We did not do this for hyenas as the female spotted hyenas feature a pseudo-scrotum which makes sexing difficult, and the visibility of males’ testes was often obscured by their large tail (Hamilton, Tilson & Frank, 1986).

In SECR modelling, σ is the scale parameter, and represents the rate of decline in the detection rate as the location of the animal’s activity centre moves away from a camera trap station. λ0 is the basal encounter rate and can be defined as the encounter rate of an animal whose activity centre lies exactly at a camera trap station. The detection rates of an individual animal decline with increasing distance between its activity centre and camera trap location (Borchers & Efford, 2008; Royle et al., 2009) and the parameter θ defines the shape of the detection function. If this parameter is estimated from the given data, the shape of the detection function could define how an animal utilizes space or resources in its environment (Elliot & Gopalaswamy, 2017). In practice, since encounter rates are so small, they are approximately equal to detection probabilities (Efford, 2019). We either used a fixed θ at 0.75 (Elliot & Gopalaswamy, 2017) and 1 (Gaussian form, Royle et al., 2009) or estimated a continuous θ parameter from the data. The complementary log-log link was used to convert encounter rates to Bernoulli detections, therefore, in our models, the probability of detecting a leopard or hyena i in pixel j is defined by a complementary log-log function of covariates.

We assessed six a priori models for leopards, and two for spotted hyenas (parameter definitions are presented in Table 1). Model 1 estimated the detection function (this is defined by θ) and assumed that detection probability is sex specific:

Table 1 Model components for secr analysis of leopard and hyena population densities.

Parameter	Definition	
n	Total number of leopards or hyenas detected during the survey period	
nz	Number of leopards augmented to n, so M = n + nz represents the maximum number of leopards in the large state space S	
σF	Rate of decline in detection probability with increasing distance between the activity center of a leopardess and the location at which female leopard was found	
σM	Rate of decline in detection probability with increasing distance between the activity center of a leopard and the location at which male leopard was found	
βsex	Difference of the complementary log-log value of detection probability between a male and female leopard	
λ0	Basal encounter rate of a leopard whose activity center is located exactly at the centroid of a grid cell	
ψ	Ratio of the true number of individuals in the population compared with the data-augmented population M	
Nsuper	Total number of leopards in the larger state space S	
ψsex	Proportion of leopards that are female (1-psi.sex/psi.sex)	
θ	Determines the shape of the estimated detection function, value θ ranges from 0.5 (exponential form) to 1 (Gaussian)	
D	Estimated density of leopards per 100 km2	

cloglog(πij)=log(λ0)+βsex−f[dist(i,j|ϑ,σsex)]

where, πij describes the detection probability on a given sampling occasion, which is a function of the basal encounter rate λ0 and distance between the activity center of individual i and pixel j, θ and sex-specific σsex. The specific form of this detection function is:

f[dist(i,j)|ϑ,σsex]=exp⌈−dist(i,j)2θ2σsex2⌉

Model 2 was based on the assumption that detection probability is not dependent on sex, (i.e., βsex was fixed at 0). The rate of decline in detection probability (σ) however, remained sex specific because this parameter is also linked to the movement of animals.

Model 3 as with model 2, had βsex set at 0 while the detection function was set at θ = 0.75

Model 4 was based on the assumption that basal encounter rate is dependent on sex, thus, βsex was fixed at 1. Rate of decline in detection probability (σ) also remained sex specific. The detection function parameter θ was fixed at 0.75.

Model 5 assumed basal encounter rate is dependent on sex but rate of decline in detection probability was independent of sex. The detection function parameter was fixed at θ = 0.75.

Model 6 was the same as model 1 but the detection function parameter (θ) was fixed at 1.

For the spotted hyenas’ density assessment, we only used model 1 and model 6 due to the lack of a sex covariate.

We used Bayesian Markov Chain Monte Carlo (MCMC) simulation and the Metropolis-Hastings algorithm (Tierney, 1994) to run our models in the package SCRbayes (https://github.com/jaroyle/SCRbayes) in the programming environment R Version 3.6.1 (R Development Core Team, 2019). We set each model to run for 20,000 iterations including a burn-in of 5,000 iterations but we adjusted this further if we did not arrive at a standing distribution, (refining burn-in period and initial iterations further). Each model was set to run for four chains (Elliot & Gopalaswamy, 2017). Model adequacy was determined by examining the Bayesian p-value on individual encounters (Royle et al., 2009). MCMC convergence was assessed using the Gelman–Rubin diagnostic (Gelman & Rubin, 1992). The five input files necessary to run these analyses and accompanying R scripts are provided in the supporting information section of this manuscript (Information S3). Although we were principally interested in estimating density, we also computed posterior mean abundance across the study area of the greater LMNP system.

Results

We recorded a total of 1,444 trap nights during the 53-day survey period. Cameras were not functional due to animal interference and battery failures for 146 trap nights, and these were not included in the SECR analysis. We recorded a total of 61 temporally independent (i.e., animals counted only once in a 24-h period) detections of leopards during our camera trap survey, and 51 spotted hyena detections (Table 2). From these we recorded 112 and 42 useable flanks for leopards and hyenas respectively (51 right hyena flanks vs 32 left flanks; nine excluded due to not identifiable or juvenile hyena). We identified 20 unique leopards (six adult males and 14 adult females), and 27 (no sex noted) spotted hyenas. This equates to a detection rate of 1.38 leopards and 1.87 spotted hyenas per 100 trap-nights.

Table 2 Number of flanks of hyenas and leopards recorded during camera trapping in LMNP with total recaptures.

Species	Number of left flanks	Number of right flanks	Number of useable flanks for analysis	Unique individuals identified	Unique individuals recaptured	
Spotted hyena	32	51	42	27	8	
Leopard	57	55	112	20	13	

Density estimates and model diagnostics

Bayesian p-values for all our leopard density models ranged from 0.61–0.76 (Table 3), indicating an adequate model fit (extremities 0.15–0.85). Convergence of models was indicated by a mean potential shrink reduction factor of <1.2 for each parameter for each model (Gelman & Rubin, 1992, Information S1). The same assessment of model adequacy was recorded for a model where sigma was estimated without a sex effect for the estimates of spotted hyena density (Bayesian p = 0.61 and shrink reduction factor for all parameters <1.2, Information S2). Model selection using marginal likelihood from Dey, Delampady & Gopalaswamy (2019) indicated that model 5, which considered basal encounter rate to be dependent on sex but detection probability independent of sex had the highest log likelihood score (log likelihood = −55,615.56, Table 3).

Table 3 SECR models from the Lake Mburo survey.

Models used to generate our density analyses for leopards and spotted hyenas in the LMNP, Uganda, 2018. We present the model number, Bayes p-value to signify model adequacy and the marginal likelihood values used to select our models, and number of iterations used to achieve convergence.

Species	Model number	Bayes p-value	Marginal likelihood	Total iterations	Burn in required to reach convergence	
Leopards	1	0.71	−62,893.814	52,000	42,000	
2	0.71	−62,885.778	50,000	20,000	
3	0.72	−62,784.534	80,000	2,000	
4	0.71	−62,729.456	50,000	20,000	
5	0.61	−55,615.556	50,000	20,000	
6	0.76	−62,985.962	50,000	20,000	
Spotted hyenas	1	0.62	−41,030.296	11,000	6,000	
2	0.64	−41,045.548	11,000	1,000	

Leopard density estimates

Using model 5, leopard density for LMNP was estimated at 6.31 individuals/100 km2 (posterior SD 1.47, 95% CI range [3.75–9.20, Table 4]). The posterior mean abundance for the Lake Mburo National Park was 24.87 (posterior SD 7.78) using this model. The leopard movement parameter or sigma σ for males and females from this model was 1.33 km (this movement parameter is a measurement of how far animals travel in the landscape and is related to home range size; Braczkowski et al., 2020b). The next best-ranked candidate model (model number 4) which considered sex as a factor affecting detection probability estimated a movement parameter of 1.60 km for males and 0.59 km for females.

Table 4 Parameter estimates with accompanying posterior standard deviation for our spatially explicit capture recapture models estimating leopard and hyena densities in the Lake Mburo National Park, Uganda.

Species	Model number	sigma (σm)	sigma2 (σf)	lam0 (λ0)	beta sex (βsex)	Psi (ψ)	 ψsex	Theta (θ)	Density (D)	
Post. Est	PSD	Post. Est	PSD	Post. Est	PSD	Post. Est	PSD	Post. Est	PSD	Post. Est	PSD	Post. Est	PSD	Post. Est	PSD	
Leopard	1	2.59	1.08	0.73	0.14	0.02	0.02	2.04	0.67	0.19	0.05	0.88	0.08	0.74	0.13	8.92	2.14	
2	2.13	1.07	0.67	0.11	0.14	0.07	0	0	0.2	0.05	0.94	0.04	0.65	0.11	9.33	2.28	
3	2.81	0.5	0.75	0.08	0.11	0.04	0	0	0.2	0.05	0.95	0.03	0.75	0	9.31	2.24	
4	4.97	3.23	0.74	0.08	0.04	0.17	1.94	0.88	0.18	0.05	0.89	0.08	0.75	0	8.82	2.14	
5	1.33	0.1	1.33	0.1	0.04	0.01	0.09	0.04	0.14	0.03	0.69	0.12	0.75	0	6.31	1.47	
6	5.09	1.76	0.97	0.12	0.02	0.01	1.94	0.66	0.19	0.05	0.88	0.08	1	0	8.83	2.11	
Spotted hyena	1	3.22	0.74	3.22	0.74	0.005	0.002	0	0	0.23	0.07	0.0003	0.0001	1	0	11.00	3.35	
2	1.9	0.64	1.9	0.64	0.007	0.003	0	0	0.24	0.08	0.0003	0.0001	0.75	0.14	11.26	3.56	

Hyena density estimates

For spotted hyenas, right flanks were recorded with the highest frequency (Table 2). Spotted hyena density for LMNP was estimated at 11.00 individuals/100 km2 (posterior SD = 0.32, 95% CI range [5.57–17.09, Table 4]) using model 1. The spotted hyena movement parameter σ for both sexes combined was 3.15 km. The posterior mean abundance for the entire state space buffer was 39.07 spotted hyenas (posterior = SD 13.51, Table 3).

Discussion

We provide a robust estimate of leopard densities and abundance in the LMNP ecosystem, southwestern Uganda, and also the first SECR assessment for spotted hyenas in Uganda as a whole (however these had wider confidence intervals when compared to leopards). These estimates are important baselines for the future monitoring of leopard and spotted hyena populations in the LMNP, one which experiences both significant levels of human-carnivore conflict and trophy hunting (Braczkowski et al., 2020c). Robust estimates of population abundance and densities are a critical cornerstone for tracking changes and trends in carnivore populations over time (e.g., Balme, Slotow & Hunter, 2009; Williams et al., 2017). In this human-carnivore conflict-prone area, it is unknown whether retaliatory killings following depredation on livestock are sustainable in the long term, especially as the LMNP is small and isolated from other larger protected areas. Previous research has shown that carnivore populations in small, isolated national parks cannot withstand the edge effects from human-carnivore conflict (e.g., from cattle farming) and trophy hunting (Woodroffe & Ginsberg, 1998; Balme, Slotow & Hunter, 2009).

Possible explanations for observed leopard and hyena densities

Leopard densities in LMNP were on the mid-tier of estimates recorded in the recent literature using SECR studies (n = 15 studies from 2013–2018, Table 5). The leopard densities we observed at 6.31 individuals/100 km2 are somewhat surprising given (a) the small size of LMNP, and (b) the high levels of conflict between these two carnivores and the livestock rearing communities on the park edge (Braczkowski et al., 2020c). Contrastingly, the hyena densities were similar to a SECR study in uMkhuze Game Reserve, northern Kwa-Zulu Natal, South Africa (a savanna system) which estimated a density of 10.59 individuals/100 km2 (posterior SD = 2.10, De Blocq, 2014), and a study in Botswana’s Moremi estimated 11.80 (posterior SD = 2.60, Rich et al., 2019). We postulate that three factors may be contributing to these densities, namely (1) the availability of preferred prey, (2) the existence of a compensation scheme that reimburses ranchers after depredation events on the LMNP edge (Braczkowski et al., 2020c), and (3) the functional extinction of lions in the region dating back to over a decade ago (at the time of publication there was only one male lion (≥10 years old) in this ecosystem, a vagrant thought to have come from Akagera National Park, in neighbouring Rwanda). LMNP is one of only two protected areas in Uganda with a population of impala, the most preferred prey of leopards (Hayward et al., 2006). The most recent studies implemented using distance sampling by Rannestad et al. (2006) and Kisame et al. (2018) found significant populations of impala within LMNP and on the adjacent cattle farmlands at 25.6 ± 4.8 individuals/km2 in the 2003 study of Rannestad et al. (2006), and 15.3 and 16.6 individuals/km2 in the 2014 and 2016 sampling periods of Kisame et al. (2018). Importantly Rannestad et al. (2006) also found a higher number of impala groups (80 vs 58) and total individuals (348 vs 255) in the community lands adjoining the park than within the national park in the wet season of 2003. Similarly, Kisame et al. (2018) estimated that nearly half of the impala population in the LMNP and surrounding ranches was found on non-protected land. Other densities of key leopard prey species estimated in this study included 3.8 ± 0.8 individuals/km2 for bushbuck (higher densities outside) and warthogs (12.3 ± 2.9 individuals/km2, densities lower outside national park, Rannestad et al., 2006). The availability of these species at relatively high densities both inside and beyond the edge of LMNP could be one reason for the densities of leopards and hyenas we observed in our study. It also remains unclear whether the functional extinction of lions in the LMNP has contributed to some level of release of leopards and spotted hyenas. For example, from their study of leopard densities in three Kwazulu-Natal Parks, South Africa, Ramesh et al. (2017) found that where lion distribution overlapped spatially with leopards, densities of leopards decreased drastically. However this pattern of leopard suppression by lions was not observed in the Sabi Sand Game Reserve, a protected area system adjacent to the Kruger National Park where leopard-lion observations have been recorded since the 1970’s (Balme et al., 2017b).

Table 5 Literature review of recent SECR leopard studies.

A review of 17 recent SECR studies performed on leopards in the last 5 years from the peer-reviewed literature. Some studies used a combination of maximum likelihood and Bayesian-based modelling approaches and therefore contain two or more estimates. We excluded the following studies for the following reasons: Goswami & Ganesh (2014)–no error reporting around estimates Kittle, Watson & Fernando (2017)–SECR results of tracks places results in contention Rich et al. (2019)–estimate is not directly reported only a figure is present. *We examined the first 10 pages of Google Scholar and limited the studies in this table to (a) those using SECR and (b) being published in the last 5 years.

Study name	Location	Habitat type	Model used to estimate density	Density estimate (leopards/100 km2)	SD (SE)	
Balme et al. (2019)	Sabi-Sands Game Reserve, South Africa	Semi-wooded savanna	Borchers & Efford (2008)	11.80	2.60	
Borah et al. (2014)	Manas National Park, India	Tropical forest and mountains	Borchers & Efford (2008)	3.40	0.82	
Braczkowski et al. (2016)	Phinda Private Game Reserve, South Africa	Savanna	Royle et al. (2009)	3.55	1.04	
Braczkowski et al. (2016)	Phinda Private Game Reserve, South Africa	Savanna	Borchers & Efford (2008)	3.40	1.20	
Devens et al. (2018)	Baviaanskloof mountains, South Africa	Mountain fynbos and forest	Royle et al. (2009)	0.24	0.10	
Devens et al. (2018)	Langeberg mountains, South Africa	Mountain fynbos and forest	Royle et al. (2009)	1.89	0.30	
du Preez, Loveridge & Macdonald (2014)	Bubye Valley Conservancy, Zimbabwe	Mopane woodland (savanna)	Borchers & Efford (2008)	5.28	0.89	
du Preez, Loveridge & Macdonald (2014)	Bubye Valley Conservancy, Zimbabwe	Mopane woodland (savanna)	Royle et al. (2009)	5.46	1.14	
Hedges et al. (2015)	Kenyir Wildlife Corridor, Malaysia	Dipterocarp forest	Borchers & Efford (2008)	3.30	1.28	
Hedges et al. (2015)	Kenyir Wildlife Corridor, Malaysia	Dipterocarp forest	Efford (2011)	3.06	0.91	
Kittle, Watson & Fernando (2017)	Horton Plains, Sri-Lanka	Montane forest	Borchers & Efford (2008)	13.40	6.3	
Ngoprasert, Lynam & Gale (2017)	Ban Krang, Kaeng Krachan National Park, Thailand	Evergreen forest	Borchers & Efford (2008)	2.50	1.20	
Qi et al. (2015)	Laoye mountains, China	Deciduous forest	Royle et al. (2009)	0.62	0.15	
Rahman et al. (2018)	Ujong Kulon National Park, Java, Indonesia	Tropical forest	Borchers & Efford (2008)	12.80	1.99	
Rahman et al. (2018)	Ujong Kulon National Park, Java, Indonesia	Tropical forest	Royle et al. (2009)	11.54	1.22	
Ramesh et al. (2017)	Ndumo Game Reserve, South Africa	Woodland savanna	Royle et al. (2009)	1.60	–	
Ramesh et al. (2017)	Western Shores, South Africa	Coastal savanna	Royle et al. (2009)	8.40	–	
Rostro-García et al. (2018)	Srepok wildlife sanctuary, Cambodia	Dry deciduous forest	Royle et al. (2009)	1.00	0.40	
Selvan et al. (2014)	Pakke Tiger Reserve, India	Tropical forest	Borchers & Efford (2008)	2.82	1.20	
Strampelli et al. (2018)	Xonghile Game Reserve, Mozambique	Woodlands and thickets (savanna)	Borchers & Efford (2008)	2.59	0.96	
Swanepoel, Somers & Dalerum (2015)	Farming matrix, Waterberg, South Africa	Livestock and game farms	Borchers & Efford (2008)	6.59	5.20	
Swanepoel, Somers & Dalerum (2015)	Lapalala Game Reserve, South Africa	Mountain bushveld (dystrophic savanna)	Borchers & Efford (2008)	5.35	2.93	
Swanepoel, Somers & Dalerum (2015)	Welgevonden Game Reserve, South Africa	Mountain bushveld (dystrophic savanna)	Borchers & Efford (2008)	4.56	1.35	
Shrestha et al. (2014)	Parsa Wildlife Reserve, Nepal	Dry deciduous forest	Efford (2004)	3.78	0.85	
Shrestha et al. (2014)	Parsa Wildlife Reserve, Nepal	Dry deciduous forest	Royle et al. (2009)	3.48	0.83	
Williams et al., 2017	Soutpansberg mountains, South Africa	Matrix of livestock farms, nature reserves, mountains	Royle et al. (2009)	5.34	0.02	

Spotted hyenas and lions have an intricate relationship of facilitation and competition (Périquet, Fritz & Revilla, 2015). Unlike leopards, spotted hyenas do not show a negative correlation with lion presence in Africa (Périquet, Fritz & Revilla, 2015) despite intraguild predation and the negative impact that lions can have on hyena reproduction (Watts & Holekamp, 2008). Spotted hyenas may benefit from the presence of lions–and vice versa–due to the high dietary overlap between the species leading to scavenging and kleptoparasitic opportunities (Hayward, 2006; Davidson et al., 2019). Observed positive correlations in lion and spotted hyena density in many parts of Africa may also be a result of their similar preferred prey base. In Zambia, M’soka et al. (2016) found a high density of spotted hyenas in a lion-depleted ecosystem, though it was suggested that the observed density was driven by the availability of wildebeest, as in Höner et al. (2005). The spotted hyena densities we observed in our study were similar to an unpublished SECR study from uMkhuze Game Reserve, KwaZulu-Natal, South Africa (De Blocq, 2014), and a study from Botswana’s Moremi (Rich et al., 2019). Estimates of spotted hyena densities using non-SECR methods, from African savanna sites range widely from 2–20 individuals/100 km2 in the Kruger National Park, South Africa (Mills, Juritz & Zucchini, 2001) to over 100 individuals/100 km2 in the Ngorongoro Crater, Tanzania (Kruuk, 1972; Höner et al., 2005). The spotted hyena density from this study is similar those from protected areas in southern Africa but lower than those in other East African savannas (Holekamp & Dloniak, 2010). It is important to note that the majority of previous estimates have been produced using non-spatial methods (e.g., call-ups and mark-resight), and to our knowledge our study is one of the first to use a SECR approach for spotted hyena density estimation (Table 6). SECR densities are typically lower for large carnivores due to other methods making more generalized extrapolations over a given unit area (Noss et al., 2012) which may explain the difference between our results and those from other savanna systems in East Africa where non-spatial methods were used.

Table 6 Spotted hyena densities recorded in the literature.

Spotted hyena density estimates using SECR and camera trapping in six locations across sub-Saharan Africa.

Study name	Location	Habitat type	Model used to estimate density	Density estimate (hyenas/100 km2)	SD (SE)	
Vissia, Wadhwa & van Langevelde (2021)	Central Tuli, Botswana	Riverine woodland and shrub savanna	Borchers & Efford (2008)	14.90	2.23	
Rich et al. (2019)	Moremi Game Reserve and cattle matrix, Botswana	Semi-wooded savanna	Borchers & Efford (2008)	11.80	2.60	
Briers-Louw (2017)	Majete Game Reserve, Malawi	Tropical dry woodland/miombo savanna woodland	Royle et al. (2009)	2.69	0.48	
De Blocq (2014)	uMhkhuze Game Reserve, South Africa	Semi-wooded savanna	Royle et al. (2009)	10.59	2.1	
O’Brien & Kinnaird (2011)	Mpala Ranch, Kenya	Semi-wooded savanna/cattle ranch	Borchers & Efford (2008)	4.93	1.7	

It is noteworthy that areas of high density between the species do not appear to overlap (Fig. 3). Previous studies have suggested that spotted hyenas can be significant kleptoparasites of leopard kills, forcing them to cache or avoid areas with high hyena density (Balme et al., 2017a; Davis et al., 2021). Similarly, another study detected low temporal overlap between leopards and spotted hyenas in Tanzania, which was postulated to be due to the avoidance of kleptoparasitism (Havmøller et al., 2020). Therefore, the avoidance of kleptoparastism may drive the differences in space use between the species we detected in LMNP but would require further investigation.

Figure 3 Densities of leopards and hyenas in Lake Mburo.

African leopard and spotted hyena detection frequencies (denoted in frequency by the size of spheres) and density estimates from our SECR models, LMNP, Uganda.

Limitations and future monitoring of large carnivores in LMNP

Our study is limited by a lack of temporal replication. This is important as we could not generate critical population parameters such as emigration, immigration, birth and death (e.g., Karanth et al., 2006). These parameters are indicators of population trend and are ultimately required to ascertain the true trajectory of a given population. It should also be remembered that spotted hyenas live in fission-fusion clans and may move together in groups or singularly. It remains to be seen if this clan-living structure may cause biases in estimates of density and other parameters in our sampling situation. For example, López-Bao et al. (2018) show that wolf densities are not significantly affected by group living. Similarly, Bischof et al. (2020) suggest that if there are low to moderate levels of gregariousness observed in group living individuals, there is little overdispersion that occurs in the estimation of the detection function and scale parameter. However, if gregariousness is high, overdispersion may be observed in confidence intervals around parameter estimates, affecting the veracity of estimates. Although our study represents the first snapshot of this leopard and spotted hyena population, it is important as a baseline estimate from which future estimates can be made against (e.g., Balme, Slotow & Hunter, 2009). Our study also failed to quantify any relationships between hyenas and leopards, which in some sites have been shown to positively influence one another’s occupancy in a landscape (Comley et al., 2020).

There is a growing conflict between large carnivores and humans in the greater LMNP ecosystem (Braczkowski et al., 2020c). The impacts of spotted hyenas and leopards on cattle, sheep and goats in the Bahima pastoral lands adjacent to LMNP are significant, and leopards and spotted hyenas were the source of 98% (n = 1,102) of depredation events recorded between January 2009–December 2018 in the region (Braczkowski et al., 2020c). Other studies have highlighted spotted hyenas as a primary source of livestock loss, which combined with their negative public image, makes them vulnerable to retaliatory killing (Kissui, 2008; Holmern, Nyahongo & Røskaft, 2007). While spotted hyenas are behaviourally flexible, populations are slow to recover following even moderate reduction (Benhaiem et al., 2018). This pattern has also been observed for African leopards (e.g., Balme, Slotow & Hunter, 2009). For this reason, the continued monitoring of the LMNP spotted hyena and leopard population is crucial from a human-carnivore conflict perspective. Continued population monitoring of leopards is also critical in the context of trophy-hunting of leopard and leopard prey, which is allowed on properties adjoining the LMNP. Even though legal harvests of leopards in Uganda since 2007 have been low (17 skins, skulls and trophies exported from 2009–2017), and 28 leopards are available on quota country-wide annually (Braczkowski et al., 2015b), it is critical to monitor these populations annually or biannually as they can rapidly decline under even modest harvest pressures (Balme, Slotow & Hunter, 2010). The way in which quotas have been set in Uganda for leopards was also done using a non-robust method which related rainfall to leopard densities (CITES CoP 14 Proposal 3).

Conclusion

We aimed at providing the first leopard and spotted hyena population density estimates for the Lake Mburo ecosystem in Uganda, a small but regionally important national park with significant cattle farming on its edge. We found that leopard occur at a relatively high density of 6.3 individuals/100 km2, probably due to a combination of factors such a high local prey density and an absence of lions. Spotted hyena densities were also relatively high, with several factors putatively at play, including abundance of prey including livestock, the absence of lions, and the general tolerance of hyenas for human disturbance. Our estimates form a robust baseline for future population monitoring to inform both the design of sustainable management offtakes, and conservation interventions for the two species in the region.

Supplemental Information

Supplemental Information 1 Hyena captures in Lake Mburo survey.

Click here for additional data file.

Supplemental Information 2 Hyena sex information for secr analysis.

Click here for additional data file.

Supplemental Information 3 Sex information for African leopards captured in the survey.

Click here for additional data file.

Supplemental Information 4 Leopard captures in Lake Mburo camera trap survey.

Click here for additional data file.

Supplemental Information 5 Habitat file for secr analysis.

Click here for additional data file.

Supplemental Information 6 Code to run all secr analyses for LMNP surveys of hyena and leopard.

Click here for additional data file.

Supplemental Information 7 Trap locations Lake Mburo camera trap survey.

Click here for additional data file.

Supplemental Information 8 Traps only no functionality.

Click here for additional data file.

We are grateful to the Ugandan Wildlife Authority for their support in the implementation of this study, particularly Aggrey Rwetsiba and Kule Asa Musinguzi.

Additional Information and Declarations

Competing Interests

Author Contributions

Animal Ethics

Field Study Permissions

Data Availability

Ralph Schenk, Nicholas Swanson and Merlin Swanson are employed by the Mihingo Lodge.

Aleksander Braczkowski conceived and designed the experiments, performed the experiments, analyzed the data, prepared figures and/or tables, authored or reviewed drafts of the paper, and approved the final draft.

Ralph Schenk conceived and designed the experiments, performed the experiments, authored or reviewed drafts of the paper, and approved the final draft.

Dinal Samarasinghe performed the experiments, analyzed the data, prepared figures and/or tables, authored or reviewed drafts of the paper, and approved the final draft.

Duan Biggs analyzed the data, authored or reviewed drafts of the paper, and approved the final draft.

Allie Richardson performed the experiments, analyzed the data, authored or reviewed drafts of the paper, and approved the final draft.

Nicholas Swanson conceived and designed the experiments, performed the experiments, authored or reviewed drafts of the paper, and approved the final draft.

Merlin Swanson conceived and designed the experiments, performed the experiments, authored or reviewed drafts of the paper, and approved the final draft.

Arjun Dheer performed the experiments, analyzed the data, authored or reviewed drafts of the paper, and approved the final draft.

Julien Fattebert conceived and designed the experiments, performed the experiments, analyzed the data, authored or reviewed drafts of the paper, and approved the final draft.

The following information was supplied relating to ethical approvals (i.e., approving body and any reference numbers):

This was a camera trap study and did not involve either the handling, capture or immobilisation of any vertebrates.

The following information was supplied relating to field study approvals (i.e., approving body and any reference numbers):

This research was granted approval by the Uganda Wildlife Authority under permit number: UWA/COD/96/05 as approved by the Executive Director Mr. Stephen Masaba.

The following information was supplied regarding data availability:

All data and code are available in the Supplemental Files.

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
