# Peer review of "Leopard and spotted hyena densities in the Lake Mburo National Park, southwestern Uganda"

_PeerJ, doi:10.7717/peerj.12307_

## Round 0.1 · original submission · Major Revisions

We have received two in depth reviews (see below) and both reviewers really like your study, but have added a number of comments and questions. Please address each of the reviewer comments and submit a revised version of your manuscripts when you are ready. Thanks very much!

·

Basic reporting

This is a clear and concise manuscrit, easily followed and understood.

Experimental design

The methods used are adequate for the purpose here.
I would appreciate some details about camera trap placements as well as why no cameras were deployed in the western part of the park.

Validity of the findings

Analyses are straightforward and the result adequately presented.
The quality of the discussion is a bit poor but can easily be improved.

Additional comments

Dear authors,

You present a simple and straightforward ms, not testing any hypothesis but providing first estimated for spotted hyaena and leopard densities in Uganda. It that sense, your work contributes to the global knowledge about these 2 species and is using recognized methods to estimate densities and population size. The language is clear and the ms manuscript well written, easily understandable. The statistical methods employed are also adequate and properly used.

However, I would recommend that you provide additional details in the methods in terms of data collections and SECR models descriptions. Supporting materials also require some work to be truly of value for readers.

I have 2 main concerns.
The first one is about camera trap placement as in the choice of the actual location where cameras were deployed in the landscape and why no cameras have been deployed in the western part of the park (Fig 1). Can you explain why it was not done?
Second, the authors claim in the discussion that one weakness of their work is that it provides no information about the relation between leopards and hyaenas. However, figure 3 shows variation in local density for both species, so possible so correlative analyses would be possible, and I would recommend it to be added to the ms.

Finally, I am somewhat disappointed by the quality of the discussion that often strays for form the actual focus of this paper (eg compensation schemes for livestock losses). I feel the quality of the discussion can be greatly improved and its length shorten to better fit such a concise aim as this work presents. For instance, you never discuss of the variation in local densities shown in Fig 3. Would you have some explanation for such variations?

You will find detailed comments on the attached pdf of your manuscript.

The supporting material 2 & 3 require some work to make a more valuable contribution, especially in terms of formatting. I would recommend to also add an R script, using one of the models as an example so reader could repeat your analyses. This was, you also would not have to provide the lengthy details in the way you did. But should you decide to provide them anyway, I would recommend formatting them in proper table instead of simple copy-paste from the R console.
I also provided comments on these 2 files, all combined in a single pdf with the MS.

I sincerely hope these comments will be helpful.

·

Basic reporting

I have no comments on the language of the manuscript. It is well written and professional.
There are gaps in the litterature cited and the literature list and tables that needs to be throughly reviewed.
The raw data is shared.
There is no hypothesis.

Experimental design

The manuscript is within the aims and scope of the journal.
There are no research questions defined, but there easily could be and the research is very important for the species and the areas it investigates.
The sampling design appears adequate and there is no ethical concerns.

Validity of the findings

Findings are mainly reporting on densities of leopard and spotted hyenas but no actual investigations into what affects their densities and the conclusions are mainly speculative albeit with revised analyses the authors would be able to make scientific inference.

Additional comments

Comments to the authors

Dear Authors of

I enjoyed reading your manuscript and commend you for your efforts. It is so little science that comes out on large carnivores in parts of East Africa and particularly Uganda. My impression is that you have a good dataset, but I do have a few concerns regarding it and some suggestions for improving it.
Currently your manuscript is a reporting on densities of leopards and spotted hyenas in Lake Mburo National Park, but there is no clear hypothesis stated nor is there any direct conservation implication and finally the use of SECR on a social canid is controversial.

Major comments
Firstly, you have good data and have a conservation approach to your study, but you do not use your data to address conservation issue. SECR is a powerful method not only for estimating densities but also to understand which factors affect the density. It is not a difficult process to add covariates on e.g., distance to national park boundary, a water source, prey density (which you are referring a lot to and must have data for given the maps you provide) to test a hypothesis on what affects densities of leopards and spotted hyenas. This will give you a scientific foundation for making conservation recommendations rather than simply speculate/postulations about them. I suggest you read the study by Havmøller et al. 2019 (which is in your references but not quoted in the text or table…) and Rather et al. 2021 DOI: 10.7717/peerj.10634 for inspiration and insight and use similar metrics and rerun your analyses with questions such as do large carnivore densities decline close to NP boundaries/villages? Are densities affected by distance to water? habitat type? prey densities? Etc.
This would set the foundation for a more insightful manuscript/publication that you can actually use as the scientific foundation for conservation recommendation – as opposed to just speculations.
Secondly, when you create a state space you should only include an area that realistically can include your study area. Right now, you are reporting and drawing conclusion on an area 20x your study site. You should use an appropriate state space and rework your entire discussion according to your new results.
Thirdly, spotted hyenas are known pack-living and occasionally pack-hunting canids. Using traditional SECR with minor changes to estimate densities is not an appropriate method because individuals do not have independent activity centres and thus densities are overestimated. There are very few studied published on SECR densities on social canids, but please read the suggested publications below and reconsider your analysis for the spotted hyenas density estimation. As your manuscript is currently composed, I am convinced that you have overestimated the hyena population.

Ngoprasert, Dusit, George A. Gale, and Andrew J. Tyre. "Abundance estimation from multiple data types for group-living animals: An example using dhole (Cuon alpinus)." Global Ecology and Conservation 20 (2019): e00792.

Harihar, Abishek, Mousumi Ghosh, Merwyn Fernandes, Bivash Pandav, and Surendra P. Goyal. "Use of photographic capture-recapture sampling to estimate density of Striped Hyena (Hyaena hyaena): implications for conservation." Mammalia 74, no. 1 (2010): 83-87.

Bohm, Torsten. "Population ecology, conservation status and genetics of the spotted hyena (Crocuta crocuta) in the Odzala-Kokoua National Park, Republic of Congo, including an assessment of the status of spotted hyenas in southeast Gabon." PhD diss., 2015.

I am suggesting major revisions to the editor and hope that you find my suggestions useful.
I look forward to reading your revised manuscript and I think it is incredibly important to have these types of studies out from this part of the world - and you have good data so you can really make analysis that can have a valuable conservation impact! You can do this!

Best regards

Rasmus Havmøller

Minor comments
Abstract: The word functionally extinct implies that lions are present, but in so low numbers that they have little or no impact on the ecosystem. Consider whether this is what you actually want to say.

L78: Uganda is a country not a state
L91: comma missing in 1585
L96: eg. Abbrivation incorrect and autocorrected capital letter subsequent
L101: should be e.g. no eg.
L107: why is the scientific name suddenly in parentheses? Check journal format requirement.
L168: female leopard home-ranges have been reported far smaller than 30km2, what is relevant to present is the inter camera trap distance, which gives the reader an idea of how many camera traps you have within a male/female home-range.
L184: half way. It is technically a result that the right flanks were captured most often, I suggest moving it to the results section.
L265: You have not defined what constitutes “temporally independent” in the methods
L268: write juvenile rather than young
L273: Bayesian p-values??? In a Bayesian framework you can only really talk about posterior probabilities to my knowledge, please just stick to that.
L290: a state-space of 6733km2?!? That is an enormous area nearly 20x your study area – is this a typo? Your math is off, 6.31 leopards / 100km2, LMNP is only 370km2, meaning 6.31x370km2 = 23.347 leopards for the entire LMNP. You are reporting +204 leopards for the national park plus 25km buffer? If your statespace needed to be this high it is an indication to me that indication that your models were a poor fit. If you have not performed a sensitivity test of your buffer, then I suggest you do so and choose the lowest possible to add realism to your study.
L294: Same as above, the math is off. If you have a density of ~11 hyenas per 100km2, and your study area/national park is 370km2, then you only have 11x3.7= 40.7 hyenas for the entire national park.
L322: But you have not tested for an effect of prey even though you could easily have done it.

---

## Round 0.2 · Minor Revisions

Both reviewers have requested further changes to your manuscript. Reviewer 2 in particular has pointed out that they were hoping you would expand on your SECR analysis. They make a very good point and I am hoping that you strongly consider adding these changes to your manuscript.

·

Basic reporting

no comment

Experimental design

no comment

Validity of the findings

no comment

Additional comments

Dear authors,

Thank you for the detailed response to my previous comments and the revised manuscript that is now much improved.

I still feel that the methods section regarding SCR and your model parameters needs some clarifications, maybe a table similar to table 3 to explain each parameter might be helpful?
While I appreciate that you split the discussion into sub-section, it does not really work as you did it. Having sections called Possible reasons for observed leopard and hyena densities and Spotted hyena densities in LMNP and comparisons to other studies is more confusing than anything else. Either change the titles to meaningful ones 'possible reasons for observed densities' is not exactly clear… or omit them.
Your table 2 still introduce parameters such as Nsuper and ψ that are never mentioned elsewhere, since detailed about the models outputs are not given. Consider deleting them from the table if you never mention them…

I attached an annotated version of the ms pdf wit specific comments.

Apart from that, I do not have further comments on your manuscript and commend you for your work.

·

Basic reporting

Language is clear, but still with some type-os and insufficiencies as per journal standards.
References are generally fine, but there are error in the reference list and some articles that are cited in main text but not present in reference list and vice versa.
Structure is professional and data shared
No hypothesis

Experimental design

Basic reporting
Basic questions that fills a knowledge gap
Rigorous analysis for leopards, more questionable but acceptable for spotted hyenas
Methods well described

Validity of the findings

Very important study with data from an area that has very little research coming out.
Data analysis is correctly performed, but does not allow for in-depth discussion which makes the discussion and conclusion very speculative

Additional comments

Dear Authors of Leopard and spotted hyena densities in the Lake Mburo National Park, southwestern Uganda

Thank you for the opportunity to once again review your manuscript. From my perspective the manuscript is improved, and I see that you have incorporated large parts of comments provided by Stéphanie and myself.
However, your lack of willingness to expand your SECR analyses to go beyond simple density reporting creates a dilemma.
You spend a lot of effort discussing that human conflict is on the increase around the national park, and then also state in your rebuttal that a visual inspection of the carnivore densities shows no sign of edge effects. This type of non-analytical subjective statements is not good scientific practice, you need the statistics to make these sorts of statements. The same goes for your discussion which is now just speculations, but could easily be a discussion of real results. Furthermore, say that you or someone else in five years repeats your study and find that carnivore densities have changed, but given that you have not investigated which factors impact carnivore densities, then you cannot conclude what has cause the decline. Was it an edge effect? Did prey decline? Did the surrounding human impact increase?
Stating that using SECR on spotted hyenas is becoming increasing commonplace in your rebuttal, is not supported in your manuscript given that the only article that you cite regarding SECR on spotted hyenas is the M’soka et al. 2016, who included a variety of parameters to understand which factors affected hyena densities (Lindsey et al. 2019 is not in your reference list……..). Your elaboration in the rebuttal on spotted hyena ecology undermines your argument for using SECR. Social animals who reside together, but might not forage together, still violates a crucial assumption in the SECR framework which is independent activity centers. The fission-fusion element of hyena ecology further complicates the use of SECR because home-ranges vary. I am all for providing the best possible density estimates of spotted hyenas, through SECR too, but it is imperative that the limitations in using SECR on this species is clearly stated and it is emphasized that further development in the line of e.g., Dusit Ngoprasert´s work is needed for more accuracy.


Minor comments

L35: check capital in species name
L39: should be “have” not “has”
L145: “,” after “e.g.”
L146: country and state might be interchangeable, but for a US reader its confusing as it sounds like Uganda is a state in the country of East Africa. Suggest you rephrase this.
L152: Crocuta Crocuta should be Crocuta crocuta according to nomenclature standards species names are never capital but genus names are.
L159: Double-space after full stop
L169: “,” after “e.g.”
L175: same as lines 35 and 152
L195-197: () missing around scientific names as per journal standards
L212: same as 195-197
L284: same as L169
L383: as above
L426: as above
L505: Lindsey et al. 2019 not in reference list
L597: comma missing after e.g.
L728: error in reference list
L747: missing indent
L801: (still) not cited in main text

---

## Round 0.3 · accepted · Accept

Thank you for addressing all reviewer comments in this revised version of the manuscript. I am of the opinion that you addressed reviewer comments in a satisfactory way and I now recommend that your manuscript be accepted for publication.